# Machine Learning Approaches to Predict Chronic Lower Back Pain in People Aged over 50 Years

**DOI:** 10.3390/medicina57111230

**Published:** 2021-11-11

**Authors:** Jae-Geum Shim, Kyoung-Ho Ryu, Eun-Ah Cho, Jin Hee Ahn, Hong Kyoon Kim, Yoon-Ju Lee, Sung Hyun Lee

**Affiliations:** Department of Anesthesiology and Pain Medicine, Kangbuk Samsung Hospital, Sungkyunkwan University School of Medicine, Seoul 03181, Korea; jgshim77@naver.com (J.-G.S.); kyoungho.ryu@hotmail.com (K.-H.R.); eunah.cho@daum.net (E.-A.C.); blatt.ajh@gmail.com (J.H.A.); mujukdol@naver.com (H.K.K.); childduck89@naver.com (Y.-J.L.)

**Keywords:** chronic lower back pain, machine learning, artificial neural network, logistic regression k-nearest neighbors, naïve Bayes, decision tree, random forest, gradient boosting machine, support vector machine, prediction

## Abstract

*Background and Objectives*: Chronic lower back pain (LBP) is a common clinical disorder. The early identification of patients who will develop chronic LBP would help develop preventive measures and treatment. We aimed to develop machine learning models that can accurately predict the risk of chronic LBP. *Materials and Methods*: Data from the Sixth Korea National Health and Nutrition Examination Survey conducted in 2014 and 2015 (KNHANES VI-2, 3) were screened for selecting patients with chronic LBP. LBP lasting >30 days in the past 3 months was defined as chronic LBP in the survey. The following classification models with machine learning algorithms were developed and validated to predict chronic LBP: logistic regression (LR), k-nearest neighbors (KNN), naïve Bayes (NB), decision tree (DT), random forest (RF), gradient boosting machine (GBM), support vector machine (SVM), and artificial neural network (ANN). The performance of these models was compared with respect to the area under the receiver operating characteristic curve (AUROC). *Results*: A total of 6119 patients were analyzed in this study, of which 1394 had LBP. The feature selected data consisted of 13 variables. The LR, KNN, NB, DT, RF, GBM, SVM, and ANN models showed performances (in terms of AUROCs) of 0.656, 0.656, 0.712, 0.671, 0.699, 0.660, 0.707, and 0.716, respectively, with ten-fold cross-validation. *Conclusions*: In this study, the ANN model was identified as the best machine learning classification model for predicting the occurrence of chronic LBP. Therefore, machine learning could be effectively applied in the identification of populations at high risk of chronic LBP.

## 1. Introduction

Lower back pain (LBP) is one of the most common musculoskeletal disorders experienced by people of all ages [1]. Around 60–80% of the general population experiences LBP at least once in their lifetime in the United States [2,3]. Globally, LBP is the leading cause of years lived with disability, which had increased substantially in the Global Burden of Disease, Injuries, and Risk Factors Study 2017 [4,5]. This causes significant personal and social losses in terms of reduced productivity and increased costs of health care [6]. Although most acute LBP patients recover well within a few weeks or months, approximately one-quarter of the patients who present to primary care settings develop chronic LBP (pain lasting for >3 months) [7]. Therefore, an understanding of the risk factors of chronic LBP and the population with a potential for chronic LBP development can help in identifying people who are at a high risk of LBP and implementing suitable preventive or treatment measures.

Machine learning is a scientific discipline that uses computer algorithms to identify patterns in large amounts of data, which can also be used to make predictions based on novel datasets [8]. Machine learning has shown excellent performance in improving the predictive value of statistics in medical imaging and postoperative clinical outcomes [9,10,11,12,13]. Although there have been studies in the past attempting to predict LBP risk, previous research had limitations, such as only applying the Cox proportional-hazards model or not incorporating psychosocial factors and ergonomics-related variables [14].

Currently, there is no existing research on models that predict the occurrence of LBP using machine learning. Therefore, we undertook this study to develop and validate a selection of machine learning models to construct an LBP predictor.

## 2. Materials and Methods

### 2.1. Data Collection

Data from respondents who participated in the Korea National Health and Nutrition Examination Surveys (KNHANES) VI-2 and VI-3 (2014–2015) were retrospectively analyzed. The KNHANES is a nationwide, cross-sectional study conducted annually by the Korea Centers for Disease Control and Prevention using a nationwide, multistage, stratified, clustered, random sampling method [15]. It evaluates demographic and clinical data, including those of sex, geographic information, and age, in the Korean population [16]. In this study, data were collected to assess the health and nutritional status of Koreans. Individuals under 50 years of age were excluded because the KNHANES IV-2 and IV-3 did not evaluate or provide data related to LBP in this age group. Therefore, 6119 respondents who participated in the chronic LBP examination survey, aged 50–89 years, were included in the study.

### 2.2. Clinical Data and Outcomes

We collected data on all patients’ demographic and clinical characteristics from the KNHANES IV-2 and IV-3. Twenty-five predictor variables were collected and used in our proposed models. Patient demographic variables included age, sex, body mass index (BMI), occupation, education level, household income, and marital status. Comorbidity variables included hypertension, diabetes mellitus, hyperlipidemia, ischemic heart disease, cerebrovascular disease, osteoarthritis, and rheumatoid arthritis. Psychosocial variables included depression symptoms, stress, sleep duration, smoking status, and alcohol intake status. We also collected data on sitting time, physical activity, fasting blood glucose levels, and chronic LBP were defined by a simple survey response to a question regarding experiencing LBP lasting >30 days in the past 3 months. Sitting time was divided into two categories: >7 h and <7 h based on the median (7 h). Physical activity was divided into two categories based on the response “yes” to the question: “Does your job involve medium-intensity physical activity that lasts for at least 10 min or makes your heart beats slightly faster?”. Stress was divided into two categories based on responses to the question “How much stress do you feel in your daily life?”. Smoking status and alcohol intake status were divided into two categories, depending on whether the participants usually smoke or drink.

### 2.3. Statistical Analyses

R software version 3.6.1 (R Development Core Team, Vienna, Austria) was used for the analysis. The following packages for machine learning were used: Caret (https://CRAN.R-project.org/package=caret, accessed on 10 September 2021), Xgboost (https://CRAN.R-project.org/package=xgboost, accessed on 11 August 2021), and Keras (https://CRAN.R-project.org/package=keras, accessed on 11 August 2021). The Caret package was used for logistic regression, k-nearest neighbor, naïve Bayes, decision tree, random forest, and support vector machine. The Xgboost package was used for gradient boosting machine. The Keras package was used for the artificial neural network (ANN). The entire code of our machine learning algorithm (https://github.com/jgshim/chronicLBP, accessed on 11 August 2021) is freely available.

Before constructing the machine learning models, our collected data were randomly segregated into training and test sets. Specifically, 70% of the data was used for training the prediction models, and 30% was used as the test set for verification. A 10-fold cross-validation approach was used to choose a set of optimal hyperparameters. The missing data were estimated using a nearest neighbor imputation algorithm, which is a similarity-based method to fill in missing data that relies on distance metrics [17]. The synthetic minority oversampling technique, addressing imbalanced datasets, was used to oversample the minority classes to overcome the low incidence of chronic LBP in the training set [18].

We identified 25 potential features, including demographic and clinical variables from previous studies conducted to identify features that may potentially affect LBP risk. Feature selection is the process of selecting features that contributes the most to the output prediction for an efficient functioning of the machine learning algorithms [19,20]. In this process, recursive feature elimination (RFE) was used as a wrapper-type feature selection algorithm to help select features. RFE works by fitting the random forest function from the Caret package in the core of the model, ranking features by importance, and removing the least important features; a specified number of features remains, as seen in Figure 1. To construct the machine learning model, we included only a subset of the available features resulting from RFE.

Model performance evaluation was conducted using the area under the receiver operating characteristic curve (AUROC), accuracy, sensitivity, and specificity. The AUROC from each machine learning model was plotted using the test dataset as a strong indicator of performance for classifiers in imbalanced datasets [21,22]. Although our data come from a nationwide study, nested cross-validation was used to estimate an unbiased generalization performance in addition to simple cross-validation. Nested cross-validation consists of a double loop. An inner loop serves for parameter selection over the validation set by fitting a model to each training set. The outer layer will be used for estimating the generalization error by averaging the test set scores over several dataset splits.

### 2.4. Ethics Statement

The VI-2 and VI-3 versions of the KNHANES were approved by the Institutional Review Board of the Korea Centers for Disease Control and Prevention (approval no. 2013-12EXP-03-5C and 2015-01-02-6C) and complied with the Declaration of Helsinki. Each participant voluntarily provided written informed consent before participating in this study. Additionally, the Institutional Review Board of Kangbuk Samsung Hospital waived the need for approval because the KNHANES survey data are openly published (approval no. KBSMC 2020-07-001).

## 3. Results

### 3.1. Patients’ Characteristics

We analyzed the data of 6119 patients who participated in the KNHANES IV-2 and IV-3 from 1 January 2014 to 31 December 2015. A total of 1394 patients (22.8%) experienced chronic LBP. The demographic and patient characteristics of the complete dataset are summarized in Table 1.

### 3.2. Feature Selection

The input variables after RFE included age, sex, BMI, household income, diabetes mellitus, hyperlipidemia, ischemic heart disease, osteoarthritis, depression symptoms, smoking status, physical activity, sitting time, and fasting blood glucose levels. The 13 features following final feature selection were used as input variables in creating the machine learning models for predicting the occurrence of chronic LBP. Correlation analyses showed a weak positive correlation between age, osteoarthritis, and chronic LBP (Figure 2).

### 3.3. Model Performance

After applying the test dataset for all machine learning techniques for predicting chronic LBP, the AUROCs calculated were 0.656 (95% CI, 0.634–0.678) for logistic regression, 0.656 (95% CI, 0.628–0.685) for k-nearest neighbor, 0.712 (95% CI, 0.685–0.740) for naïve Bayes, 0.671 (95% CI, 0.643–0.698) for decision tree, 0.699 (95% CI, 0.671–0.728) for random forest, 0.660 (95% CI, 0.631–0.690) for gradient boosting machines, 0.707 (95% CI, 0.678–0.735) for support vector machine, and 0.716 (95% CI, 0.689–0.744) for ANN, as seen in Table 2. The ANN method achieved the best performance in terms of AUROCs, as well as accuracy, sensitivity, and specificity (Figure 3). Results of nested cross validation are shown in Table 3.

## 4. Discussion

Previous reports have highlighted a lack of well-validated models for predicting LBP [23]. Well-verified risk prediction models help in the identification of patients at a high risk of disease and help in the implementation of preventive measures in advance. The objective of this study was to demonstrate that machine learning algorithms could accurately predict the occurrence of chronic LBP.

Feature selection is an important concept in machine learning, especially when dealing with a dataset that contains numerous features. This type of dataset is referred to as a high-dimensional dataset, with a multitude of problems, including a long training time for a machine learning model. The objective of feature selection is to improve the prediction performance of predictors and aid better understanding of the underlying principle in the dataset. In our study, analysis and modeling with RFE facilitated the identification of patients at a high risk of LBP and the determination of clinical factors associated with chronic LBP. A previous study employed the Cox proportional-hazards model to identify patients at a high risk of LBP [14,24]. However, only one model’s performance was obtained, and it was impossible to compare the various models. Another previous study applied stepwise logistic regression analysis to predict whether a patient with a recent new episode of LBP would develop persistent pain [25]. Stepwise methods have well-known limitations, such as unstable variable selection and biased coefficient estimation. In this study, we developed and validated our models by performing feature selection, cross-validation, and testing using different machine learning algorithms. Thus, we anticipate that the effective implementation of machine learning methods in clinical settings may facilitate the provision of personalized medicine to patients with chronic LBP in the future.

The handling of missing data is a major concern in machine learning and different application domains, including medical areas. In this study, we applied the nearest neighbor imputation algorithm for extrapolating the missing data rather than deletion. However, different methods exists for imputing missing data. Recently, oversampling methods have been proposed to impute missing data or generate valid synthetic instances to train classifiers in the case of extreme scarcity of training data. Izonin et al. showed the high accurate prediction using data augmentation procedure and support vector regression [26]. Additionally, Salazar et al. proposed a new method using generative adversarial networks and vector Markov random field to effectively improve the classifier performance [27].

Each machine learning algorithm has its own hyperparameters, such as the number of hidden layers in ANN or number of features available for splitting at each tree node in a random forest [11]. It is a parameter that is set before the learning process begins. In our study, we found that the optimal ANN, specifically multilayer perceptron, was composed of two hidden layers to predict the occurrence of chronic LBP. In the ANN model, the first and second hidden layers included 20 and 10 nodes, respectively, which were interconnected. Since the most optimal hyperparameters should often be specified by the researcher or set using heuristics to construct the ANN model, we obtained the suitable hyperparameters empirically, as seen in Appendix A. The hyperparameters found in this study could be useful for further research using the ANN method.

This study had certain limitations. The study used data from a cross-sectional survey that involved looking at data from a population at one specific point in time. Thus, it is not guaranteed to be representative, and the temporal relationship between predictor variables and chronic LBP cannot be determined. In addition, the prediction model in our study was based on a Korean population that is over 50 years of age. Thus, it may be difficult to generalize our study to different age groups considering the unique characteristics of Korean culture, such as sitting posture and high-intensity working hours. Clinically, it is meaningful to show an accuracy of 71.7% in prediction, but it still requires further research. It is doubtful that demographic data and clinical information are enough to accurately predict chronic LBP. Model alterations are most likely necessary for better predictive model. One possible region of our interest includes the lumbar spine X-ray, computed tomography, or magnetic resonance imaging.

## 5. Conclusions

This study is important because it promotes the identification of patients at high risk of chronic LBP in a population of Koreans over 50 years of age using machine learning. Among the machine learning models that were developed and validated, the ANN model was found to be the best machine learning classification model for predicting the occurrence of chronic LBP.

## Figures and Tables

**Figure 1 medicina-57-01230-f001:**
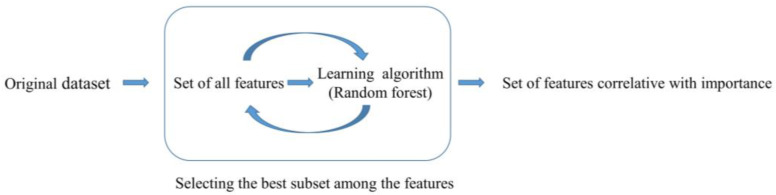
Schematic representation of recursive feature elimination (RFE) in random forest algorithm.

**Figure 2 medicina-57-01230-f002:**
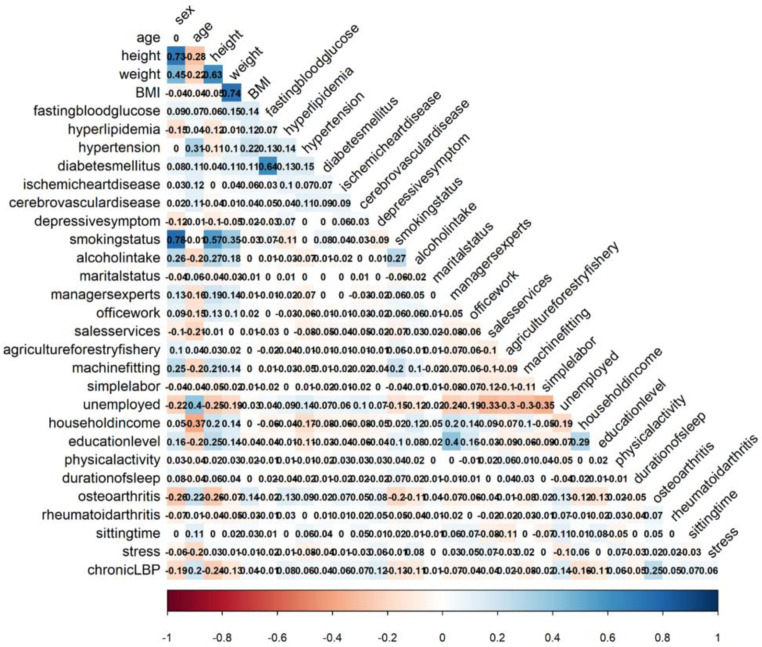
Correlation between variables.

**Figure 3 medicina-57-01230-f003:**
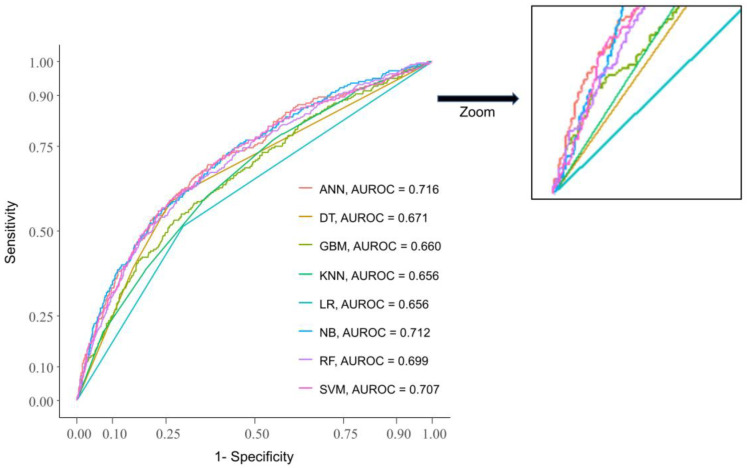
Areas under the receiver operating curve for test data.

**Table 1 medicina-57-01230-t001:** Demographic data and variable features of the included population.

Variables	All Cases (*n* = 6119)	No Lower Back Pain (*n* = 4725)	Lower Back Pain (*n* = 1394)	*p*-Value
Age (years)	64 (56–72)	62 (56–70)	69 (60–76)	<0.001
Sex (female)	3511 (57.4%)	2464 (52.1%)	1047 (75.1%)	<0.001
BMI (kg/cm^2^)	23.9 (22.0–26.0)	23.9 (22.0–25.9)	24.1 (21.9–26.4)	0.001
Comorbidities (*n*)				
Hypertension	3006 (49.1%)	2249 (47.6%)	757 (54.3%)	<0.001
Diabetes mellitus	1020 (16.7%)	747 (15.8%)	273 (19.6%)	<0.001
Hyperlipidemia	1449 (23.7%)	1027 (21.7%)	422 (30.3%)	<0.001
Ischemic heart disease	280 (4.6%)	182 (3.8%)	98 (7.0%)	<0.001
Cerebrovascular accident	253 (4.1%)	161 (3.4%)	92 (6.6%)	<0.001
Osteoarthritis	1294 (21.1%)	736 (15.6%)	558 (40.0%)	<0.001
Rheumatoid arthritis	163 (2.7%)	106 (2.2%)	57 (4.1%)	<0.001
Education (*n*)	878 (14.3%)	773 (16.4%)	105 (7.5%)	<0.001
Marital status (*n*)	6045 (98.8%)	4666 (98.8%)	1379 (98.9%)	0.70
Household income (*n*)	2636 (43.1%)	2238 (47.4%)	398 (28.6%)	<0.001
Occupation (*n*)				<0.001
Managers, experts	330 (5.4%)	295 (6.2%)	35 (2.5%)	
Office work	213 (3.5%)	185 (3.9%)	28 (2.0%)	
Sales and services	599 (9.8%)	490 (10.4%)	109 (7.8%)	
Agriculture, forestry, and fishery	493 (8.1%)	370 (7.8%)	123 (8.8%)	
Machine fitting	509 (8.3%)	448 (9.5%)	61 (4.4%)	
Simple labor	672 (11.0%)	531 (11.2%)	141 (10.1%)	
Unemployed (student, housewife, etc.)	3303 (54.0%)	2406 (50.9%)	897 (64.3%)	
Sitting time (*n*)	2845 (46.5%)	2110 (44.7%)	735 (52.7%)	<0.001
Duration of sleep (*n*)	3210 (52.5%)	2548 (53.9%)	662 (47.5%)	<0.001
Smoking (*n*)	2402 (39.3%)	2022 (42.8%)	380 (27.3%)	<0.001
Alcohol intake (*n*)	4940 (80.7%)	3928 (83.1%)	1012 (72.6%)	<0.001
Depressive symptom (*n*)	364 (6.0%)	206 (4.4%)	158 (11.3%)	<0.001
Stress (*n*)	4633 (75.7%)	3515 (74.4%)	1118 (80.2%)	<0.001
Physical activity (*n*)	437 (7.1%)	297 (6.3%)	140 (10.0%)	<0.001
Fasting blood glucose (mg/dL)	99 (92–110)	99 (92–110)	99 (92–109)	0.69

KNHANES, The Korea National Health and Nutrition Examination Survey; BMI, body mass index. The data are presented as medians (interquartile ranges) or numbers (%).

**Table 2 medicina-57-01230-t002:** Performance of all machine learning models.

Model	AUROC (95% CI)	Accuracy (95% CI)	Sensitivity (95% CI)	Specificity (95% CI)
LR	0.656 (0.634–0.678)	0.608 (0.582–0.634)	0.82 (0.79–0.84)	0.36 (0.32–0.40)
KNN	0.656 (0.628–0.685)	0.631 (0.608–0.653)	0.83 (0.81–0.85)	0.35 (0.32–0.39)
NB	0.712 (0.685–0.740)	0.713 (0.692–0.733)	0.84 (0.82–0.86)	0.43 (0.39–0.47)
DT	0.671 (0.643–0.698)	0.665 (0.643–0.687)	0.85 (0.83–0.87)	0.39 (0.35–0.42)
RF	0.699 (0.671–0.728)	0.701 (0.680–0.722)	0.84 (0.81–0.86)	0.42 (0.38–0.46)
GBM	0.660 (0.631–0.690)	0.689 (0.667- 0.710)	0.82 (0.80–0.84)	0.39 (0.35–0.43)
SVM	0.707 (0.678–0.735)	0.677 (0.656–0.699)	0.85 (0.83–0.87)	0.40 (0.36–0.44)
ANN	0.716 (0.689–0.744)	0.717 (0.696–0.734)	0.84 (0.82–0.86)	0.44 (0.40–0.48)

AUROC, area under the receiver operating characteristic curve; CI, confidence interval; LR, logistic regression; KNN, k-nearest neighbors; NB, naïve Bayes; DT, decision tree; RF, random forest; GBM, gradient boosting machine; SVM support vector machine; ANN, artificial neural network.

**Table 3 medicina-57-01230-t003:** Nested cross validation results of all machine learning models.

Model	AUROC (k = 1)	AUROC (k = 2)	AUROC (k = 3)	AUROC (k = 4)	AUROC (k = 5)	AUROC (mean + SD)
LR	0.690	0.607	0.679	0.637	0.651	0.653 ± 0.033
KNN	0.612	0.676	0.604	0.626	0.579	0.619 ± 0.036
NB	0.610	0.649	0.602	0.671	0.671	0.641 ± 0.033
DT	0.636	0.710	0.579	0.669	0.597	0.638 ± 0.053
RF	0.654	0.714	0.677	0.633	0.636	0.663 ± 0.034
GBM	0.538	0.612	0.661	0.637	0.628	0.615 ± 0.047
SVM	0.700	0.665	0.674	0.726	0.691	0.691 ± 0.024
ANN	0.728	0.718	0.739	0.662	0.724	0.714 ± 0.030

AUROC, area under the receiver operating characteristic curve; LR, logistic regression; KNN, k-nearest neighbors; NB, naïve Bayes; DT, decision tree; RF, random forest; GBM, gradient boosting machine; SVM support vector machine; ANN, artificial neural network; k, number of folds in the outer loop of nested cross-validation; SD, standard deviation.

## Data Availability

The data used to support the findings of this study are available from the corresponding author upon request.

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
