# Peer review of "Machine Learning Approaches to Predict Chronic Lower Back Pain in People Aged over 50 Years"

_medicina, 2021, doi:10.3390/medicina57111230_

Round 1

Reviewer 1 Report

The quality of the paper has been improved significantly. All my concerns have been adequately addressed in the revised version of the paper including improvement of literal presentation; extended explanation of the implementation details; rationale of the selection and tuning of the methods; improvement of result discussion; and outline of future research. Therefore, I consider the paper should be ready for publication after final editing revision.

Reviewer 2 Report

No further comments from my side. All review comments were addressed.

This manuscript is a resubmission of an earlier submission. The following is a list of the peer review reports and author responses from that submission.

Round 1

Reviewer 1 Report

In this paper, a ML analysis has been conducted from 8 classifiers towards predicting chronic lower back pain in people aged over 50 years. An extensive cohort has been used and the manuscript is well-written. However, further clarification/editing is required. In particular:

1) please place each of the references in front of the dots (end of the sentence)

2) page 2/9, Currently, There

3) You can skip this, too generic: "It comes from the intersection of statistics and computer science to learn relationships from data using efficient computing algorithms.[9]" 

4) According  to your code in github it seems that preprocessing has been implemented on the entire set before train/test split. This includes bias in the model training phase since histogram parameters (i.e. for center, scaling) are obtained from the entire set. Please elaborate on that.

5) It is not clear to me if feature selection from RFE has been performed on the overall or just the training set. Please comment on this.

6) Since your dataset contained non-numeric data, did you consider this when you applied the nearest neighbor imputation algorithm which imputes based on distances? From your code I see that all non-numeric variables were dichotomous. If yes, NN is indeed acceptable for your variables. Please comment.

7) I would expect at least one external validation set, since this is a multicentric study, or an iterative train/test split (inner-outer cross-validation) to assess generalization in model's performance.   

Reviewer 2 Report

The main objective of the paper is an experimental study of several machine learning methods applied to classify patients of chronic lower back pain (LBP). Several classifiers such as logistic regression (LR), k-nearest neighbors (KNN), naïve Bayes (NB), decision tree (DT), random forest (RF), gradient boosting machine (GBM), support vector machine (SVM), and artificial neural network (ANN) and feature selection using recursive feature elimination (RFE) were implemented. All these methods are well known, and thus, possible contribution of the paper would be focused on experimental findings. The contribution of the paper is fair considering that the presented results are interesting from a practical standpoint. However, several issues in the paper should be resolved. Literal presentation of the paper has room for improvement. Considering the results obtained, several details of the methods applied as well as discussion on alternative methods need more explanation. Lines of future research should be specified. In summary, I consider the contents of the paper are potentially publishable, but the following specific issues should be addressed in a revised version of the paper.

 - Literal presentation of the paper has room for improvement. For instance, (i) Page 3, please enlarge font size of text in Figure 1. (ii) Page 3, the last two paragraphs are disconnected. Last sentence of the first of these paragraphs is cut. (iii) Table 1, showing the selected features, should be referred in third paragraph of page 3. (iv) Figure 2 is unreadable. (v) Please check for definition of acronyms, e.g., “CI”. (vi) Page 6, Discussion section, “LBP. [24]” or “LBP [24].”? (vii) It is difficult to distinguish the curves in Figure 3. Please try to better identify these curves, particularly for reading in white and black printing. A zoom in the area of interest of low values for 1-specifity could help.

Therefore a proofreading of the paper is recommended.

- Please specify what is the type of neural network that was implemented (multilayer perceptron?). Besides, include a rationale of the selection of that method. Nowadays, there is an increasing interest in convolutional neural networks (CNN), see for instance reference [23].

- The dataset used contained missing data that were filled in using a nearest neighbor imputation algorithm. Oversampling methods provides a competitive solution to this problem. They can be used for imputation of missing fields in the data record or to generate synthetic data record samples that avoid possible bias of classifiers due to class data imbalance. Thus, providing smoothed version of the estimators. Recently, oversampling methods were introduced to handle the training with very small training datasets. Please discuss about alternatives for data imputation. I suggest the following references, https://doi.org/10.3390/sym13040612; https://doi.org/10.1016/j.eswa.2020.113819.

- Recursive feature elimination (RFE) was selected for a feature selection processing step. Please explain the motivation for using this method in comparison with methods of dimension reduction such as principal component analysis (PCA) that has been successfully applied in several classification problems.

- Page 6, “The objective of this study was to demonstrate that machine learning algorithms could accurately predict the occurrence of chronic LBP.” The highest result of classification accuracy obtained was around 70%. Please explain if this can be considered an accurate prediction for chronic LBP.

- Page 7, “… with a multitude of problems, including a long training time for machine learning model and overfitting.” I understand, overfitting was solved by making the training and testing datasets disjoint. Thus, “and overfitting” should be eliminate from that sentence.

- Please add outline of future lines of research including improvements to the proposed method.
